# Impact of Lymphocyte and Neutrophil Counts on Mortality Risk in Severe Community-Acquired Pneumonia with or without Septic Shock

**DOI:** 10.3390/jcm8050754

**Published:** 2019-05-27

**Authors:** Estel Güell, Marta Martín-Fernandez, Mari C. De la Torre, Elisabet Palomera, Mateu Serra, Rafael Martinez, Manel Solsona, Gloria Miró, Jordi Vallès, Samuel Fernández, Edgar Cortés, Vanessa Ferrer, Marc Morales, Juan C. Yébenes, Jordi Almirall, Jesús F. Bermejo-Martin

**Affiliations:** 1Department of Intensive Care Medicine, Hospital de Mataró, Universitat Autònoma de Barcelona, 08304 Barcelona, Spain; mctorre@csdm.cat (M.C.D.l.T.); epalomera@csdm.cat (E.P.); mserra@csdm.cat (M.S.); rmartinez@csdm.cat (R.M.); msolsona@csdm.cat (M.S.); gmiro@csdm.cat (G.M.); jcyebenes@csdm.cat (J.C.Y.); jalmirall@csdm.cat (J.A.); 2Laboratory of Biomedical Research in Sepsis (BioSepsis), Hospital Clínico Universitario de Valladolid, Instituto de Investigación Biomédica de Salamanca (IBSAL), 37007 Salamanca, Spain; mmartin.iecscyl@saludcastillayleon.es (M.M.-F.); jfbermejo@saludcastillayleon.es (J.F.B.-M.); 3Intensive Care Medicine, Hospital Parc Taulí-Sabadell, 08208 Sabadell, Spain; jvalles@tauli.cat (J.V.); sfernandezv@tauli.cat (S.F.); ecortess@tauli.cat (E.C.); vferrer@tauli.cat (V.F.); marcmoralescodina@gmail.com (M.M.); 4Centro de Investigación Biomedica En Red-Enfermedades Respiratorias (CibeRes, CB06/06/0028), Instituto de salud Carlos III (ISCIII), Av. de Monforte de Lemos, 5, 28029 Madrid, Spain; 5Grup d’Estudi al Maresme de la Pneumònia Adquirida a la Comunitat (GEMPAC) acreditat per la AGAUR (expedient 2014 SGR 1018), Consorci Sanitari del Maresme, 08204 Mataró, Spain; 6Programa de Doctorat en Medicina de la Universitat Autònoma de Barcelona, Passeig de la Vall d’Hebrón 119-129, 08035 Barcelona, Spain

**Keywords:** community-acquired pneumonia, mortality risk, leukocyte subclasses counts, neutrophils, lymphocytes

## Abstract

Background: Community-acquired pneumonia (CAP) is a frequent cause of death worldwide. As recently described, CAP shows different biological endotypes. Improving characterization of these endotypes is needed to optimize individualized treatment of this disease. The potential value of the leukogram to assist prognosis in severe CAP has not been previously addressed. Methods: A cohort of 710 patients with CAP admitted to the intensive care units (ICUs) at Hospital of Mataró and Parc Taulí Hospital of Sabadell was retrospectively analyzed. Patients were split in those with septic shock (*n* = 304) and those with no septic shock (*n* = 406). A single blood sample was drawn from all the patients at the time of admission to the emergency room. ICU mortality was the main outcome. Results: Multivariate analysis demonstrated that lymphopenia <675 cells/mm^3^ or <501 cells/mm^3^ translated into 2.32- and 3.76-fold risk of mortality in patients with or without septic shock, respectively. In turn, neutrophil counts were associated with prognosis just in the group of patients with septic shock, where neutrophils <8850 cells/mm^3^ translated into 3.6-fold risk of mortality. Conclusion: lymphopenia is a preserved risk factor for mortality across the different clinical presentations of severe CAP (sCAP), while failing to expand circulating neutrophils counts beyond the upper limit of normality represents an incremental immunological failure observed just in those patients with the most severe form of CAP, septic shock.

## 1. Introduction

Community-acquired pneumonia (CAP) with an incidence of 1.62 cases per 1000 adults/year is a frequent cause of death worldwide. As many as 10% to 22% of the patients suffer from severe forms of the disease requiring critical care [1,2,3]. Mortality remains unacceptably high despite advances in critical care management, ranging from 21% to 58% in those admitted in the intensive care unit (ICU) [4,5]. Delayed admission to an ICU is associated with increased hospital length of stay and higher ICU mortality for patients hospitalized with CAP [6,7]. Moreover, Renaud et al. showed that some patients not fulfilling the major criteria for severe CAP according to the Infectious Diseases Society of America/American Thoracic Society consensus guidelines may still benefit from early transfer to the ICU [8,9].

The precise mechanisms of lung cell injury in pneumonia caused by various pathogens remain unknown; pathogen-related substances (including toxins or pathogen-associated molecular patterns (PAMPs)) or substances from injured host cells or immune cells by infectious insults (including proinflammatory cytokines and damage-associated molecular patterns (DAMPs)) may induce host immune reactions, and this may be responsible for lung cell injury [10]. As recently described, CAP shows different biological endotypes [11]. Improving characterization of these endotypes is needed to optimize individualized treatment of this disease. Immunological profiling is a helpful tool to do so [12]. Evaluating the key elements of the innate and adaptive immunity in patients with CAP helps to assess severity and prognosis in this disease [13].

The leukogram has been demonstrated to be useful to identity those patients with severe infections suffering from complicated outcomes. For example, septic shock in patients who fail to expand circulating neutrophil counts in blood (CNC) present an increased risk of mortality [14]. Lymphopenia accompanied by a rise in neutrophil count are events commonly observed in bacteremia [15,16]. In a cohort of patients with ventilator-associated pneumonia (VAP) caused by *S. aureus*, Rodriguez-Fernandez et al. demonstrated that total eosinophil counts at VAP diagnosis were a protective factor against mortality in the first 28 days [17]. In CAP, Curbelo et al. showed that the percentage of neutrophils is a better predictor of mortality in the early-stage evolution than PSI score and CURB65 scores [18]. Marrie et al. determined those factors that predict in-hospital mortality of patients admitted with CAP, and lymphopenia was associated with early mortality [19]. As we have recently described, patients with CAP needing hospitalization with less than 724 lymphocytes/mm^3^ show a 1.93-fold increment in the risk of mortality, independently of the CURB-65 score, critical illness, and receiving an appropriate antibiotic treatment [20]. 

In addition to their individual values, the ratio between neutrophils and lymphocytes has been evaluated with prognostic purposes. Jager et al. confirmed in a study in an emergency care setting that both lymphopenia and the neutrophil–lymphocyte count ratio (NLR) were better predictors of bacteremia than routine parameters like C-reactive protein (CRP) level. In another study, they showed that the use of NLR in patients with CAP predicts severity and outcome with a higher prognostic accuracy as compared with traditional markers, and may allow the clinician to stratify patients into different prognostic categories [21,22]. Akilli et al. calculated the NLR in critical patients treated in the emergency department, evidencing that high ratios were associated with increased mortality [23]. Cataudella et al., in a cohort of elderly adults with CAP, confirmed that NLR predicted mortality and performed better than PSI, CURB-65, CRP, and white blood cell count [24].

Although the prognostic ability of neutrophil and lymphocyte counts has been studied in different kinds of infections, surprisingly its specific value for prognosis assessment in critically ill patients with severe CAP (sCAP) has not been previously addressed. The objective of the present study was to evaluate the impact of neutrophil and lymphocyte counts on the risk of mortality in sCAP depending on the presence or absence of septic shock.

## 2. Materials and Methods

### 2.1. Study Design and Patient Selection

This was a retrospective analysis of a cohort of 710 patients admitted to the ICUs of the “Hospital of Mataró” and “Parc Taulí Hospital of Sabadell” with a diagnosis of CAP between January 1999 and March 2016. They were followed up until ICU discharge. 

We evaluated the association between leukocyte subclasses counts at admission to the emergency room and the risk of ICU mortality. 

Inclusion criteria included age ≥18 years old, presence of severe CAP as cause of admission to the ICU. CAP was defined by the presence of acute lower respiratory tract infection symptoms with the appearance of focal signs on physical examination of the chest and new radiological findings suggestive of a pulmonary infiltrate in chest radiograph. Only those patients showing a complete leukogram in the first 24 h following admission to the emergency room were considered in the analysis. Septic shock was defined according to the definition proposed by the Society of Critical Care Medicine (SCCM), The European Society of Intensive Care Medicine (ESICM), The American College of Chest Physicians (ACCP), the American Thoracic Society (ATS), and the Surgical Infection Society (SIS) in 2001 [25]. Following this definition, septic shock in adults refers to a state of acute circulatory failure (infection-related) characterized by persistent arterial hypotension unexplained by other causes. Hypotension is defined by a systolic arterial pressure below 90 mmHg, mean arterial pressure lower than 60, or a reduction in systolic blood pressure of more than 40 mmHg from baseline, despite adequate volume resuscitation, in the absence of other causes of hypotension.

Exclusion criteria included nosocomial pneumonia, healthcare-associated pneumonia (HCAP), and presence of immunosuppression (human immunodeficiency virus infection, hematological disease, cytotoxic therapy, solid organ transplantation, corticosteroid treatment with >20mg/day of prednisone in the last three months).

### 2.2. Ethics

The Ethics Committee of the *Consorci Sanitari del Maresme* approved the study (Code: 13/11). Given the observational and retrospective nature of the study, informed consent was waived. This study fulfils the standards indicated by the Declaration of Helsinki.

### 2.3. Leukocyte Quantification

A single blood sample was drawn from all patients at the time of admission to the emergency room, and counts of the different leukocyte subtypes were assessed at the central laboratories of the participant hospitals following standard operative procedures approved for clinical use. Only neutrophil and lymphocyte counts were available in the databases analyzed in this study. 

### 2.4. Main Outcome Measures

Mortality at the ICU was the main outcome. The severity of disease at presentation was assessed by identifying the necessity of mechanical ventilation and by calculating the APACHE II score upon patient’s admission [26].

### 2.5. Other Study Variables

Demographics characteristics, comorbidities were recorded as follows: chronic respiratory disease, such as chronic bronchitis (clinically defined) or chronic obstructive respiratory disease (diagnosed using spirometry criteria), asthma, bronchiectasis or sequelae of tuberculosis or other interstitial lung disease; chronic cardiovascular disease, ischemic heart disease (clinical features or diagnostic studies), valvular heart disease, or other disorders such as rhythm disturbances or malformations; diabetes mellitus (defined as glucose intolerance or treatment with oral antidiabetic drugs or insulin); acute renal failure and chronic renal failure (defined as serum creatinine levels >2 mg/dL or blood urea nitrogen >40 mg/dL); liver disease, solid organ neoplasia; chemotherapy; HIV; liver disease; hematological neoplasia and intake of corticosteroids. Microbiological and laboratory data were also collected using a standardized datasheet. 

### 2.6. Statistical Analysis

For the demographic and clinical characteristics of the patients, differences between groups were assessed using the Chi-square test for categorical variables and the Mann–Whitney U test for continuous variables when appropriate. The ability of neutrophil and lymphocyte counts to differentiate survivors from nonsurvivors in patients with and without septic shock was initially assessed by using the area under the receiver operating characteristic curve analysis (AUC). The cut-off for neutrophil and lymphocyte counts regarding mortality prediction was obtained by calculating the optimal operating point (OOP) in the AUC, as previously described [27]. The OOP was considered the value for which the point on the curve had the minimum distance to the upper left corner (where sensitivity = 1 and specificity = 1). By Pythagoras’ theorem, this distance is:$$\mathrm{OOP}\;=\;\sqrt{{\left(1-sensitivity\right)}^{2\;}+{\left(1-specificity\right)}^{2\;}}$$

The ability of this cut-off value to predict ICU mortality was further evaluated by using multivariate logistic regression analysis. Those potential confounding variables yielding *p* values <0.2 in the univariate regression analysis were further introduced in the multivariate analysis as adjusting variables for neutrophil or lymphocyte counts. The calibration efficiency was tested using the Hosmer–Lemeshow test. Data analyses were performed using IBM SPSS Statistics version 20.0 (Armonk, New York, NY, USA). 

## 3. Results

### 3.1. Patient Characteristics (Table 1)

Patients of both groups (sCAP with septic shock and with no septic shock) showed similar characteristics in terms of age, sex, and accompanying comorbidities. As expected, patients with septic shock were more severe at ICU admission, as indicated by their higher APACHE-II score, showed a higher ICU mortality, and presented more frequently an infection by Gram+ bacteria. In contrast, patients with no septic shock showed more frequently a viral infection. In addition, 67% of the patients in the group of sCAP with septic shock and 58% in the group with no septic shock showed lymphopenia (<1000 cells/mm^3^).

### 3.2. Analysis of Mortality Risk in the Group of Patients with sCAP and Septic Shock

AUROC analysis in this group established 8850 cells/mm^3^ neutrophils and 675 lymphocytes/mm^3^ as the OOPs for identifying nonsurvivors at the ICU (Figure 1). Sensitivity and specificity for these OOPs were (69.5%/61.6%) for the neutrophil count and (60.6%/62.5%) for the lymphocyte count.

In this group, presenting at the emergency room with neutrophil and lymphocyte counts below these OOPs was independently associated with an increased risk of ICU mortality (Table 2 and Table 3). 

Goodness-of-fit *p* values for both regression models were greater than 0.05, suggesting good calibration (Hosmer–Lemeshow χ^2^
*p* = 0.306 for the model involving the neutrophil counts and *p* = 0.614 for the model involving the lymphocyte counts). 

### 3.3. Analysis of Mortality Risk in the Group of Patients with sCAP and No Septic Shock

AUROC analysis in this group established 501 lymphocytes/mm^3^ as the OOPs for identifying nonsurvivors at the ICU (Figure 2). Sensitivity and specificity for this OOP was (77.4%/51.4%).

In this group, presenting at the emergency room with lymphocyte counts <501 lymphocytes/mm^3^ was independently associated with an increased risk of ICU mortality (Table 4). 

Goodness-of-fit *p* value for the regression model was >0.05, suggesting good calibration (Hosmer–Lemeshow χ^2^
*p* = 0.252). 

## 4. Discussion

Our results evidenced that the presence of lymphopenia was a preserved predictor of mortality in sCAP patients either with or without septic shock, although the cut-off of the lymphocyte count was slightly lower in the latter. In contrast, the neutrophil count was relevant to the prognosis just in the group with septic shock. Patients failing to expand the neutrophil count over the upper limit of normality [28] at presentation to the emergency room had an increased risk of dying.

These results confirm previous findings coming from our group, reporting an association between lymphopenia and mortality risk in CAP [20]. The lymphocyte cut-offs for predicting mortality in sCAP (675 and 501 lymphocytes/mm^3^) were nonetheless lower than that found in our previous work (724 lymphocytes/mm^3^), in which we included principally patients not needing admission to the ICU. In addition, our results confirm the impact of neutrophil counts on mortality in patients with septic shock [14]. The cut-off of neutrophil counts identified to predict mortality for sCAP patients with septic shock was slightly higher (8850 cells/mm^3^) than that observed in our previous study on septic shock of any infectious origin (7226 cells/mm^3^) [14]. Identifying specific cut-offs for infections with different severity or origin is important to assist clinical decisions in an individualized manner. 

In consequence, this study reveals a different impact of lymphocyte and neutrophil counts on the prognosis of patients with sCAP. Accounting with enough circulating lymphocytes would be important to survive sCAP in every circumstance, while neutrophil count expansion would be employed by the immune system as a “last resource” mechanism restricted to the most severe form of clinical presentation of sCAP, septic shock. Our results thus suggest that the coexistence of lymphopenia and absence of circulating neutrophil count expansion would represent an incremental immunological failure in septic shock, as compared to the isolated presence of lymphopenia in those patients with no septic shock, of which mortality is 2.7-fold lower. 

Notably, 67% of the patients in the group of sCAP with septic shock and 58% in the group with no septic shock showed lymphopenia, which constitutes a surprising finding considering that patients with previous immunosuppression were excluded in our analysis. Other comorbidities such as diabetes or the role of immune senescence could explain this high prevalence of lymphopenia in severe CAP. Migration of lymphocytes to the lung or other tissues, adhesion to the endothelium, insufficient production in the bone marrow, or apoptosis during the acute phase of CAP could also contribute to explain it [29].

The leukogram is an inexpensive test that could provide complementary information to that offered by clinical scores such as PSI, CURB-65, or the PIRO system. These scores are used both for severity stratification and mortality prediction, but they have some limitations. PSI and CURB-65 overestimate weight of age, which represents a drawback for their application in young patients suffering from severe respiratory insufficiency [30]. On the other hand, laboratory parameters are mandatory for calculating CURB-65 and PSI, which are not always available in scenarios such as in Primary Care Settings. In turn, the PIRO score accurately predicts 28-day mortality in CAP patients requiring ICU admission [31]. As occurs with PSI and CURB-65, some parameters required to calculate the PIRO score, such as bacteremia, are not available in nonhospital settings. In this regard, emergence of the new Point of Care devices to obtain the leukogram will make it easier to use in these scenarios [32].

In addition, our results provide potential physio-pathological clues on the involvement of neutrophils and lymphocytes in sCAP, in a moment when therapies aimed to modulate the immune response (GM-CSF, inteleukin-7, anti-programmed death-ligand 1 drugs, corticosteroids, and/or intravenous immunoglobulin) are in the spotlight as potential therapies [29,33]. 

Finally, we assessed the value of the neutrophil-to-lymphocyte ratio (NLR) to predict mortality in the group of patients with septic shock, since neutrophil counts were not associated with mortality in the group of patients with no septic shock. Showing an NLR <12 translated to 2.55-fold risk of mortality (CI 95%: 1.41–4.60, *p* = 0.002). We also evaluated the impact of C-reactive protein (CRP) on the mortality risk, and we found no association between CRP and survival. 

Our study has an important limitation, which is the absence of quantification of the immature precursors of neutrophils in blood. Neutrophil counts used in clinical practice are those offered by the automatic analyzers, which correspond to mature neutrophils. Counting neutrophil precursors requires visual enumeration by an expert hematologist. Immature granulocytes are relevant since they are probably mediating some of the immuno-pathogenic events in sCAP [34]. Future studies should evaluate the role of myeloblasts, promyelocytes, myelocytes, metamyelocytes, band cells, and segmented neutrophilic cells in sCAP. Another limitation is the absence of information on biomarkers such as procalcitonin or midregional proadrenomedullin, which nonetheless are not available at the emergency rooms. In turn, lymphocyte subsets were not analyzed. Finally, our results should be confirmed in new cohorts using the SEPSIS-3 definition for septic shock [35]. 

## 5. Conclusions

Lymphopenia is a preserved predictor of mortality in sCAP, independent of the absence or presence of septic shock, while failing to expand neutrophil counts is a signature predicting mortality just in those patients with septic shock. These two immunological signatures could indicate the presence of incremental immunological failures in those CAP patients with the most severe form of the disease (that with higher mortality), and may be especially useful for the early identification of patients requiring ICU admission as well as for guiding their clinical management.

## Figures and Tables

**Figure 1 jcm-08-00754-f001:**
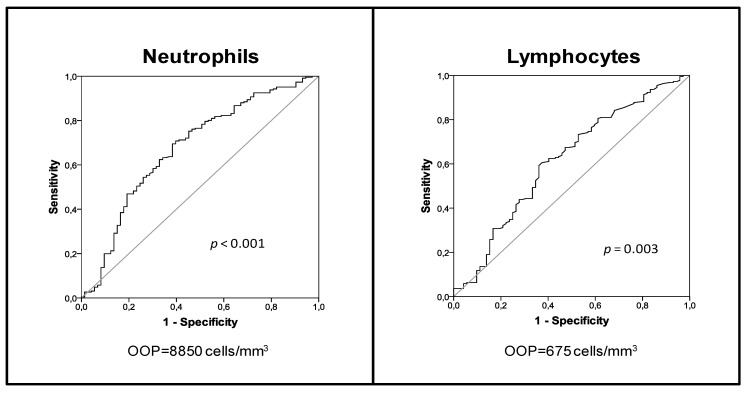
AUROC analysis for ICU mortality in sCAP patients with septic shock.

**Figure 2 jcm-08-00754-f002:**
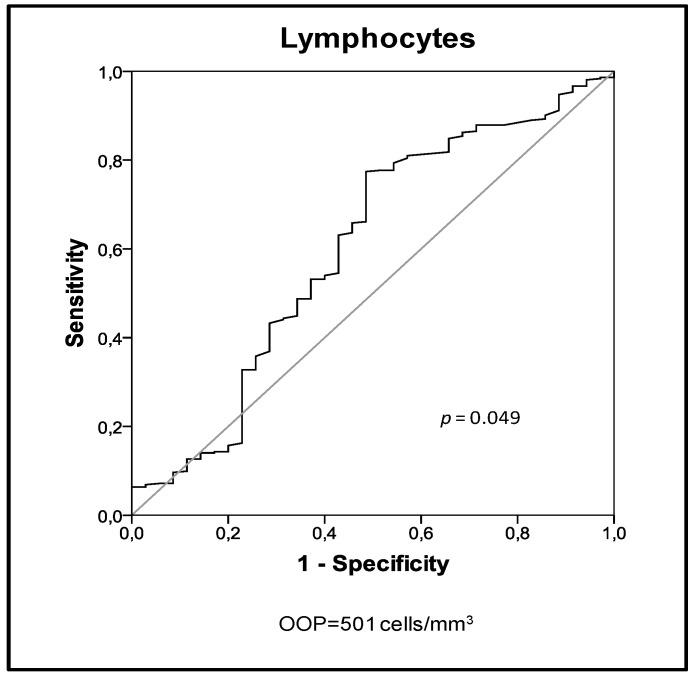
AUROC analysis for ICU mortality in sCAP patients with no septic shock.

**Table 1 jcm-08-00754-t001:** Clinical characteristics of the studied groups. n.s: not significant.

		No Septic Shock (*n* = 406)	Septic Shock (*n* = 304)	*p* Value
Characteristics	Age (years, median (IQR))	60 (30)	63 (26)	n.s.
	Male (%, (*n*))	67.0 (272)	72.4 (220)	n.s.
Comorbidities, (% (*n*))	Chronic cardiovascular disease	22.7 (92)	18.1 (55)	n.s.
	Chronic respiratory disease	33.7 (137)	37.8 (115)	n.s.
	Chronic renal failure	4.2 (17)	6.6 (20)	n.s.
	Diabetes mellitus	27.1 (110)	22.7 (69)	n.s.
Time course and outcome	Mechanical ventilation (% (*n*))	30.8 (125)	34.5 (105)	n.s.
	APACHE II (median (IQR))	15.86 (7)	18.00 (11)	<0.001
	ICU mortality (% (*n*))	9.4 (38)	25.3 (77)	<0.001
Microbiology, (% (*n*))	Gram+	33.8 (136)	57.9 (175)	<0.001
	Gram-	21.2 (85)	17.5 (53)	n.s.
	Fungi	7.0 (28)	4.3 (13)	n.s.
	Virus	14.7 (59)	7.0 (21)	0.001
	Polymicrobial	9.5 (38)	7.3 (22)	n.s.
Measurements at diagnosis, (median (IQR))	White blood cells (cells/mm^3^)	11,990 (12,745)	11,215 (18,315)	n.s.
	Lymphocytes (cells/mm^3^)	859 (895)	716 (880)	0.009
	Neutrophils (cells/mm^3^)	11,360 (10,180)	11,900 (14,350)	n.s.

**Table 2 jcm-08-00754-t002:** Logistic regression analysis for predicting ICU mortality in the sCAP group with septic shock: neutrophil count.

	OR	CI 95%		*p*
Age	1.02	1.00	1.04	0.121
APACHE-II	1.07	1.03	1.12	0.001
Gram–bacteria	2.41	1.97	6.45	<0.001
Polymicrobial infection	0.22	0.05	0.99	0.049
Neutrophils < 8850 cells/mm^3^	3.57	1.97	6.45	<0.001

**Table 3 jcm-08-00754-t003:** Logistic regression analysis for predicting ICU mortality in the sCAP group with septic shock: lymphocyte count.

	OR	CI 95%		*p*
Age	1.01	0.99	1.03	0.217
APACHE-II	1.08	1.03	1.12	<0.001
Gram–bacteria	2.04	0.91	4.56	0.083
Polymicrobial infection	0.31	0.07	1.36	0.119
Lymphocytes < 675 cells/mm^3^	2.32	1.30	4.15	0.005

**Table 4 jcm-08-00754-t004:** Logistic regression analysis for predicting ICU mortality in the sCAP group with no septic shock: lymphocyte count.

	OR	CI 95%		*p*
Age	1.02	0.99	1.04	0.278
Chronic obstructive pulmonary disease	1.53	0.69	3.36	0.294
APACHE-II	1.09	1.03	1.15	0.003
Mechanical ventilation	1.49	0.68	3.29	0.318
Gram–bacteria	1.70	0.72	4.00	0.224
Polymicrobial infection	2.90	0.93	8.99	0.066
Lymphocytes < 501 cells/mm^3^	3.76	1.74	8.14	0.001

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
