# Peer review of "Impact of Lymphocyte and Neutrophil Counts on Mortality Risk in Severe Community-Acquired Pneumonia with or without Septic Shock"

_jcm, 2019, doi:10.3390/jcm8050754_

Reviewer 1 Report

Community-acquired pneumonia (CAP) is a common and serious infection worldwide. The mortality rate of CAP in patients admitted to the ICU remains high, even in immunocompetent patients, despite antibiotics and adequate supportive care. Severity assessment is an important step in the management of CAP. Several models have been developed to predict the risk of mortality in CAP. This study aimed to evaluate the potential value of leukogram to assess the prognosis and risk of mortality in critically ill patients with severe CAP.

Similar studies were already conducted and published by the authors’ group. They demonstrated that failing to expand circulating neutrophil counts is a major prognostic factor in patients with septic shock (Crit Care Lond Engl 2014), and patients with lymphopenic CAP is associated with an increased risk of morality (EbioMedicine 2017). The identified neutrophil and lymphocyte counts are expected to be confirmed and validated in a prospective study. Therefore, overall, the novelty and creativity of this manuscript were then of less.

Specific comments:

1. Introduction. Many CAP risk prediction models have been developed to help clinicians predict pneumonia outcome and determine appropriate management more accurately, some of which are designed to predict mortality. It is suggested to descript the key characteristics and disadvantages of the existing pneumonia scores (such as PIRO, CURB-65, and PSI), and the rationales of new prediction tool in this section.

2. Materials and Methods. [2.1 Study design and patient selection]. Line 68 and Line 70. Is this a ‘prospective’ or ‘retrospective’ study? 

3. Materials and Methods. Line 71, 79 and 89. The timing of blood cell counts data available to be analyzed in this study is confusing. Was it ‘at hospital admission’, ‘admission to ICU’, or ‘admission to ER’?

4. Results. [3.2. Analysis of mortality risk]. Line 146-148. There was only one conclusive sentence in this paragraph. The authors are expected to provide data, statistics or description in details to support the results.  

5. Results. [3.3. Derivation and validation of the neutrophil cut-off value to predict 28-day mortality]. The most critical problem is the effect of confounding factor. According to authors’ previous study, septic shock in patients who failed to expand circulating neutrophil counts in blood presented an increased risk of mortality. In the present study, >40% CAP patients in both derivation and validation cohort had shock (Table 1). Therefore, shock is a potential confounder which could affect the variables (neutrophil count and mortality) and cause a spurious association in this study. It is suggested to adjust this confounder.

6. Some typos are present in the manuscript. For example, “Hongos” should be “fungus” (Table 1); “oher” should be “other” (Line 222). English editing is advised.

Author Response

Reviewer comment (RC): community-acquired pneumonia (CAP) is a common and serious infection worldwide. The mortality rate of CAP in patients admitted to the ICU remains high, even in immunocompetent patients, despite antibiotics and adequate supportive care. Severity assessment is an important step in the management of CAP. Several models have been developed to predict the risk of mortality in CAP. This study aimed to evaluate the potential value of leukogram to assess the prognosis and risk of mortality in critically ill patients with severe CAP. Similar studies were already conducted and published by the authors’ group. They demonstrated that failing to expand circulating neutrophil counts is a major prognostic factor in patients with septic shock (Crit Care Lond Engl 2014), and patients with lymphopenic CAP is associated with an increased risk of morality (EbioMedicine 2017). The identified neutrophil and lymphocyte counts are expected to be confirmed and validated in a prospective study. Therefore, overall, the novelty and creativity of this manuscript were then of less.

Author response (AR): thanks for your comments. It is true that our group is pioneer in evaluating the value of the leukogram in different context in patients with severe infections, but we strongly disagree with the reviewer`s concern on the novelty of the results presented here. In the past, we identified the cut-offs of neutrophils and lymphocytes that could inform on the prognosis of patients with septic shock or CAP. In the present work, we identified the cut-offs of neutrophils and lymphocytes that could inform on the prognosis of critically ill patients with severe CAP (sCAP). In fact, compared with CAP (PMID: 28958655), where we found that neutrophils did not influence prognosis, in sCAP neutrophil do influence prognosis, with a cut off slightly higher (8850 cells/mm3) than that observed in a previous study on septic shock in two cohorts of sepsis patients with infections of many different origins (7226 cells/mm3) (PMID: 24524810). In turn, the cut-off for lymphocytes in sCAP was lower than that found for CAP (633 cells/mm3 versus 724 cells/mm3). Identifying specific cut-offs for specific infections is important to assist clinical decisions in an individualized manner.  The discussion section has been re-edited to highlight the above mentioned aspects.

Specific comments:

RC: 1. Introduction. Many CAP risk prediction models have been developed to help clinicians predict pneumonia outcome and determine appropriate management more accurately, some of which are designed to predict mortality. It is suggested to descript the key characteristics and disadvantages of the existing pneumonia scores (such as PIRO, CURB-65, and PSI), and the rationales of new prediction tool in this section.

 AR: The aim of this study was not substituting current pneumonia scores, but to provide additional evidence on the prognostic role of a simple test such as leukogram. This provides in addition potential physiopathological clues on the involvement of neutrophils and lymphocytes in sCAP, in a moment when therapies aimed to modulate the immune response (GM-CSF. IL7. Anti-PDL1) are being proposed for treating sepsis and sCAP. PIRO, CURB-65, and PSI are discussed as requested in the new version of our article.

 RC: 2. Materials and Methods. [2.1 Study design and patient selection]. Line 68 and Line 70. Is this a ‘prospective’ or ‘retrospective’ study? 

AR: We clarified this in the new version of our article: we performed a retrospective analysis of two cohorts of patients already recruited.

RC: 3. Materials and Methods. Line 71, 79 and 89. The timing of blood cell counts data available to be analyzed in this study is confusing. Was it ‘at hospital admission’, ‘admission to ICU’, or ‘admission to ER’?

AR: we have clarified this as requested. One of the strengths of our article is that although all the patients considered here were admitted to the ICU, we have profiled leukocyte counts in the very first contact with the hospital, at the emergency room, before being admitted to the ICU.

RC:4. Results. [3.2. Analysis of mortality risk]. Line 146-148. There was only one conclusive sentence in this paragraph. The authors are expected to provide data, statistics or description in details to support the results.  

AR: this is a typo: the section 3.2. Analysis of mortality risk comprises the following sections, which are those providing the data and details:

-        3.2.1.-. Derivation and Validation of the Neutrophil Cut-off Value to predict ICU mortality

-        3.2.2.-. Derivation and Validation of the Lymphocyte Cut-off Value to predict ICU mortality

This section (with its subsections) has been correctly labelled

RC:5. Results. [3.3. Derivation and validation of the neutrophil cut-off value to predict 28-day mortality]. The most critical problem is the effect of confounding factor. According to authors’ previous study, septic shock in patients who failed to expand circulating neutrophil counts in blood presented an increased risk of mortality. In the present study, >40% CAP patients in both derivation and validation cohort had shock (Table 1). Therefore, shock is a potential confounder which could affect the variables (neutrophil count and mortality) and cause a spurious association in this study. It is suggested to adjust this confounder.

AR: the variable “septic shock” has been introduced as an adjusting variable in the multivariate models, as suggested by the reviewer, inducing no major impact in the results.

RC: 6. Some typos are present in the manuscript. For example, “Hongos” should be “fungus” (Table 1); “oher” should be “other” (Line 222). English editing is advised.

AR: we have made an effort to correct typos and improve English language throughout the manuscript.

 Reviewer 2 Report

The authors conducted an observational study regarding the relationship between the leukogram and mortality of severe community-acquired pneumonia (CAP) patients treated in the intensive care unit (ICU). The authors found that patients with low neutrophil count (< 8850/mm3) and lymphopenia (< 633/mm3) at presentation were associated with a higher mortality at 28 days, and suggested that these 2 parameters may reflect immunological signatures which identify patients with severe CAP with higher risk of mortality.

The authors performed 2 large scaled studies and demonstrated that there were statistical significances between the two immunological markers and prognosis of the disease. But for this issue, many studies already reported similar results as the authors have reviewed in this study. Thus, the authors offered rather limited new information and insufficient data regarding the results. This reviewer would like to agree with the issue of the study; fail of circulating neutrophil count expansion plus lymphopenia are associated with mortality in severe CAP. However, they analyzed only once-performed data, and the failure of expansion of neutrophil is inadequate in this condition. They should add the follow-up data for sustained or eventual lower neutrophils and lymphocytes since the immune status of patients decides progression and prognosis of the disease (CAP).

 - The immune system of patients controls not only the insults from CAP but also the insults from other diseases such as chronic organ diseases and diabetes. Thus, it is possible that when patients with comorbid diseases are affected with CAP, immune system of these patients may have a limited immune capacity because of immune consumption for comorbid diseases. In addition, severity of pneumonia of the hosts is reflected in the values of immunologic parameters, including immune cells (lymphocytes, neutrophils, and other cells) and immune proteins (CRP, cytokines, immunoglobulins, other proteins). Thus, a leukogram may be one of parameters in CAP, reflecting the severity of the diseases during clinical course of the disease, and more severe lymphopenia and possibly neutropenia with a relative neutrophilia (neutrophil to lymphocyte ratio, NLR) may be associated with higher values of some pro-inflammatory cytokines at the early stage of CAP, although the authors did not perform studies on these parameters. It is believed that during clinical course of CAP, the levels of immune cells, including lymphocytes and neutrophils are also affected by stage of the disease, severity of inflammation at presentation, and other pathologic lesions in CAP or in comorbid diseases.

In CAP as a controllable disease by immune system of the host, the host immune reaction against insults from pathogen infection is believed to be responsible for lung cell injury. If patients recover from the disease, the immune reaction of the host before the peak of inflammation (non-specific immune adaptive cells and pro-inflammatory cytokines may be involved in this stage) may be associated with tissue cell injury, and immune reaction after the peak of inflammation may be associated tissue cell repair (specific adaptive immune cells and immunoglobulins, platelets and anti-inflammatory cytokines may be involved in the convalescent stage. The intensity of systemic inflammation during this process is reflected in laboratory parameters such as white cell count with differentials, and levels of immune proteins such as CRP, LDH, cytokines and chemokines. To know the function of immunological parameters such as immune cells, cytokines or other parameters, serial examinations are needed at least in the acute stage (at presentation) and in the convalescent stage or terminal stage of the disease. Neutrophil count is a highly nonspecific parameter in all diseases and can easily be fluctuated by treatments such as corticosteroids within a day. Nonetheless, it is expected that sustained or eventual severe leucopenia may ensue in premortal patients if the cause of death is resulted from CAP, because of all immune cell consumption and impossible mobilization of immune cells to the pathologic lung lesions. Whereas in case of death caused by other organ failure, patients may show different leukogram profiles.

Thus, the authors should add more data and discuss regarding the role of immune cells including neutropenia and lymphopenia in patients with severe CAP.

 For these issues, the following contents may be helpful.

- The precise mechanisms of lung cell injury in pneumonia caused by various pathogens remain unknown; pathogen-related substances (including toxins or pathogen-associated molecular patterns (PAMPs)) or substances from injured host cells or immune cells by infectious insults (including proiflammatory cytokines and damage-associated molecular patterns (DAMPs)) may induce host immune reactions, and this may be responsible for lung cell injury (Lee KY. Infect Chemother 2015;47(1):12-26). At the molecular level, these substances have various sizes and biochemical characteristics, classifying them as protein substances and non-protein substances. Immune cells and immune proteins may recognize and act on these substances, including pathogenic proteins and peptides, depending upon the size and biochemical properties of the substances (this theory is known as the protein-homeostasis-system hypothesis). The severity or chronicity of pneumonia or ARDS depends on the amount of etiologic substances with corresponding immune reactions, the duration of the appearance of specific immune cells, or the repertoire of specific immune cells that control the substances. Therefore, treatment with early systemic immune modulators (corticosteroids and/or intravenous immunoglobulin) as soon as possible may reduce aberrant immune responses in the potential stage of ARDS (Lee KY. Int J Mol Sci 2017;18(2): E388).

-Lymphopenia or eventual leukopenia may be characteristic of severe pneumonia patients infected with respiratory pathogens, including influenza viruses, corona viruses, the measles virus, and M. pneumoniae. The severity of lymphopenia is correlated with the severity of lung injury. The autopsy findings of severe ARDS patients and experimental animals infected with influenza viruses show lymphocyte depletion of whole lymphoid tissues. This finding, together with lymphocyte predominance in early lung lesions, suggests T cells may control the substances from pathogens and/or injured host cells. Animals with depressed T cell function or loss of T cell function such as nude mice show milder or few pneumonia lesions in comparison to immune-competent animals in mycoplasma or influenza virus infection models, although the duration of pathogen detection in the lungs of animals with compromised T cells is longer. These finding suggest that lung injury is associated with T cell activation rather than with pathogens. It is possible that there is a limitation on the numerical capacity of the host immune system on mobilizing immune cells against these relentless substances to counter extensive lung cell injury in immune-competent patients. Patients with underlying diseases or immune-deficient states may have a limited repertoire of immune cells. Furthermore, severe pneumonia or ARDS from a viral infection tends to induce subsequent bacterial infections in patients, which adds to the workload of immune cells (Lee KY. Int J Mol Sci 2017;18(2): E388).

Author Response

Reviewer comment (RC): The authors conducted an observational study regarding the relationship between the leukogram and mortality of severe community-acquired pneumonia (CAP) patients treated in the intensive care unit (ICU). The authors found that patients with low neutrophil count (< 8850/mm3) and lymphopenia (< 633/mm3) at presentation were associated with a higher mortality at 28 days, and suggested that these 2 parameters may reflect immunological signatures which identify patients with severe CAP with higher risk of mortality. The authors performed 2 large scaled studies and demonstrated that there were statistical significances between the two immunological markers and prognosis of the disease. But for this issue, many studies already reported similar results as the authors have reviewed in this study. Thus, the authors offered rather limited new information and insufficient data regarding the results.

Authors’ response (AR): we agree that there are a number of studies evaluating the predictive ability of the leukogram in infection, but the present work is unique in the sense that is the first in studying this matter in two cohorts of patients with the most severe form of CAP, that deserving hospitalization to the ICU (sCAP). In addition, the cut-offs of neutrophils and lymphocytes to predict mortality vary depending on the severity and origin of the infection (see PMID: 28958655 and PMID: 24524810). In this regard, we provide here specific cut-offs of neutrophil and lymphocytes counts to predict mortality in sCAP, which could be helpful to those clinicians treating these patients in the ICU.

RC: This reviewer would like to agree with the issue of the study; fail of circulating neutrophil count expansion plus lymphopenia are associated with mortality in severe CAP. However, they analyzed only once-performed data, and the failure of expansion of neutrophil is inadequate in this condition. They should add the follow-up data for sustained or eventual lower neutrophils and lymphocytes since the immune status of patients decides progression and prognosis of the disease (CAP).

AR: We appreciate the reviewer constructive insight on our work. We understand that we have not properly explained what we refer to with the concept “failure of neutrophil count expansion”. Although normal reference values in blood vary depending on sex, race and age, available literature supports that 8550 cells/mm3 is at the upper limit of normal values for circulating neutrophil count (CNC) (PMID: 20236184). In consequence, we observe that those sCAP patients failing to expand CNC over the upper limit of normality have an increased risk of dying. This has been explained better in the new discussion section.

RC: The immune system of patients controls not only the insults from CAP but also the insults from other diseases such as chronic organ diseases and diabetes. Thus, it is possible that when patients with comorbid diseases are affected with CAP, immune system of these patients may have a limited immune capacity because of immune consumption for comorbid diseases. In addition, severity of pneumonia of the hosts is reflected in the values of immunologic parameters, including immune cells (lymphocytes, neutrophils, and other cells) and immune proteins (CRP, cytokines, immunoglobulins, other proteins). Thus, a leukogram may be one of parameters in CAP, reflecting the severity of the diseases during clinical course of the disease, and more severe lymphopenia and possibly neutropenia with a relative neutrophilia (neutrophil to lymphocyte ratio, NLR) may be associated with higher values of some pro-inflammatory cytokines at the early stage of CAP, although the authors did not perform studies on these parameters. It is believed that during clinical course of CAP, the levels of immune cells, including lymphocytes and neutrophils are also affected by stage of the disease, severity of inflammation at presentation, and other pathologic lesions in CAP or in comorbid diseases.

AR: We appreciate the reviewer comment. In fact, to control for the influence of comorbidities on the prognostic ability of lymphocytes and neutrophil counts, we used an univariate analysis to identify those accompanying conditions influencing the most prognosis. These conditions were introduced as adjusting variables in the multivariate analysis.  In addition, to evaluate the impact of inflammation on mortality in sCAP, as suggested by the reviewer, we have collected now the data on C reactive protein levels concomitantly with leukocyte counts (at admission to the emergency room) and assessed its value to predict mortality. We found no significant influence of CRP on prognosis. This information is offered now in the results section. Following the suggestions of the reviewer, in the new version of the article we assessed also the value of the neutrophil to lymphocyte ratio (NLR) to predict mortality, but we did not observe any significant association between NLR and prognosis neither in the derivation or the validation cohorts.

RC:  In CAP as a controllable disease by immune system of the host, the host immune reaction against insults from pathogen infection is believed to be responsible for lung cell injury. If patients recover from the disease, the immune reaction of the host before the peak of inflammation (non-specific immune adaptive cells and pro-inflammatory cytokines may be involved in this stage) may be associated with tissue cell injury, and immune reaction after the peak of inflammation may be associated tissue cell repair (specific adaptive immune cells and immunoglobulins, platelets and anti-inflammatory cytokines may be involved in the convalescent stage. The intensity of systemic inflammation during this process is reflected in laboratory parameters such as white cell count with differentials, and levels of immune proteins such as CRP, LDH, cytokines and chemokines. To know the function of immunological parameters such as immune cells, cytokines or other parameters, serial examinations are needed at least in the acute stage (at presentation) and in the convalescent stage or terminal stage of the disease. Neutrophil count is a highly nonspecific parameter in all diseases and can easily be fluctuated by treatments such as corticosteroids within a day. Nonetheless, it is expected that sustained or eventual severe leucopenia may ensue in premortal patients if the cause of death is resulted from CAP, because of all immune cell consumption and impossible mobilization of immune cells to the pathologic lung lesions. Whereas in case of death caused by other organ failure, patients may show different leukogram profiles. Thus, the authors should add more data and discuss regarding the role of immune cells including neutropenia and lymphopenia in patients with severe CAP.

AR: we totally agree with the reviewer that assessing just neutrophil and lymphocyte counts is a simplistic manner to approach the understanding of immunology in sCAP. But the objective of our study is not that much providing a general framework of immunopathogenesis of sCAP but providing useful cut-offs of these cells to help clinicians working at the emergency room and the ICU to make early decisions. This is the reason because our study lacks of serial examinations of the leukogram, since this was not in the initial scope of our work.  Regarding the potential influence of steroids on neutrophil count, the leukogram was profiled at the entry of the patients to the emergency room, when they had not yet receive any steroids.

RC:   For these issues, the following contents may be helpful.

- The precise mechanisms of lung cell injury in pneumonia caused by various pathogens remain unknown; pathogen-related substances (including toxins or pathogen-associated molecular patterns (PAMPs)) or substances from injured host cells or immune cells by infectious insults (including proiflammatory cytokines and damage-associated molecular patterns (DAMPs)) may induce host immune reactions, and this may be responsible for lung cell injury (Lee KY. Infect Chemother 2015;47(1):12-26). At the molecular level, these substances have various sizes and biochemical characteristics, classifying them as protein substances and non-protein substances. Immune cells and immune proteins may recognize and act on these substances, including pathogenic proteins and peptides, depending upon the size and biochemical properties of the substances (this theory is known as the protein-homeostasis-system hypothesis). The severity or chronicity of pneumonia or ARDS depends on the amount of etiologic substances with corresponding immune reactions, the duration of the appearance of specific immune cells, or the repertoire of specific immune cells that control the substances. Therefore, treatment with early systemic immune modulators (corticosteroids and/or intravenous immunoglobulin) as soon as possible may reduce aberrant immune responses in the potential stage of ARDS (Lee KY. Int J Mol Sci 2017;18(2): E388).

AR: thanks for your constructive comments. We basically agree with the reviewer on the multicausal origin of the damage in sCAP. We provide now this information in the introduction section along with the suggested references.

RC:  -Lymphopenia or eventual leukopenia may be characteristic of severe pneumonia patients infected with respiratory pathogens, including influenza viruses, corona viruses, the measles virus, and M. pneumoniae. The severity of lymphopenia is correlated with the severity of lung injury. The autopsy findings of severe ARDS patients and experimental animals infected with influenza viruses show lymphocyte depletion of whole lymphoid tissues. This finding, together with lymphocyte predominance in early lung lesions, suggests T cells may control the substances from pathogens and/or injured host cells. Animals with depressed T cell function or loss of T cell function such as nude mice show milder or few pneumonia lesions in comparison to immune-competent animals in mycoplasma or influenza virus infection models, although the duration of pathogen detection in the lungs of animals with compromised T cells is longer. These finding suggest that lung injury is associated with T cell activation rather than with pathogens. It is possible that there is a limitation on the numerical capacity of the host immune system on mobilizing immune cells against these relentless substances to counter extensive lung cell injury in immune-competent patients. Patients with underlying diseases or immune-deficient states may have a limited repertoire of immune cells. Furthermore, severe pneumonia or ARDS from a viral infection tends to induce subsequent bacterial infections in patients, which adds to the workload of immune cells (Lee KY. Int J Mol Sci 2017;18(2): E388).

AR: we explain the potential causes of lymphopenia and the role of chronic diseases in the discussion. We cite the suggested reference. Due to the translational nature of our work, which pretends to communicate clinicians that they can use lymphocyte and neutrophil counts to identify those patients with sCAP and poor outcomes, we do not enter into further explanations on the potential pathogenic implications (beneficial or detrimental) of these counts in sCAP, since our study is of observational nature and lacks of a complete immunological characterization of the patients, which was beyond the initial scope.

Round  2

Reviewer 1 Report

The authors preformed logistic regression analysis to adjust the confounding effect of septic shock. However, the better ways to exclude or control confounding variables are Randomization, Restriction and Matching. For example, all the participants were divided into two groups: sCAP with septic shock and without septic shock. It would be clear whether the neutrophil count less than 8850 cells/mm3 was a predictor of mortality mainly for those with sCAP or combined with shock.

Results. Table 1. Is the data of APACHE II in Validation cohort the ‘median’? Rechecking is suggested.

Author Response

Reviewer comment: the authors performed logistic regression analysis to adjust the confounding effect of septic shock. However, the better ways to exclude or control confounding variables are Randomization, Restriction and Matching. For example, all the participants were divided into two groups: sCAP with septic shock and without septic shock. It would be clear whether the neutrophil count less than 8850 cells/mm3 was a predictor of mortality mainly for those with sCAP or combined with shock.

Author response: the reviewer was totally right. We have now split the patients in two groups, those with no septic shock and those with septic shock. In consequence, we have completely reanalyzed the data and re-edited the methods, results and discussion section. We found now that lymphopenia was a predictor of mortality in both groups (with and without septic shock). In contrast, neutrophil counts just predicted mortality in those patients with septic shock, as the reviewer suspected. These results suggest that fail in neutrophil count expansion is an additional immunological failure to that represented by lymphopenia, restricted to the most severe form of CAP (septic shock), which has a superior mortality to that showed by those patients with no septic shock. This new version provides the new cut-offs for lymphocyte and neutrophil counts corresponding to each group (with or without septic shock). We appreciate the constructive overview of the reviewer which has translated into the identification of the proper value of lymphocyte and neutrophil counts for predicting mortality in sCAP, suggesting in addition the existence of an incremental immunological failure in septic shock.

 Reviewer comment: Results. Table 1. Is the data of APACHE II in Validation cohort the ‘median’? Rechecking is suggested.

Author response: Table 1 has been totally re-edited, as we compare now patients with or without septic shock. We do use median for APACHE II in the new table.